# Targeting Intercellular Communication in the Bone Microenvironment to Prevent Disseminated Tumor Cell Escape from Dormancy and Bone Metastatic Tumor Growth

**DOI:** 10.3390/ijms22062911

**Published:** 2021-03-13

**Authors:** Lauren M. Kreps, Christina L. Addison

**Affiliations:** 1Cancer Therapeutics Program, Ottawa Hospital Research Institute, Ottawa, ON K1H 8L6, Canada; lkreps@ohri.ca; 2Department of Biochemistry, Microbiology and Immunology, University of Ottawa, Ottawa, ON K1H 8L6, Canada; 3Department of Medicine, University of Ottawa, Ottawa, ON K1H 8L6, Canada

**Keywords:** bone metastasis, dormancy, angiogenesis, bone remodeling, immunosuppression, osteoclast, osteoblast, macrophage, MDSC, therapy

## Abstract

Metastasis to the bone is a common feature of many cancers including those of the breast, prostate, lung, thyroid and kidney. Once tumors metastasize to the bone, they are essentially incurable. Bone metastasis is a complex process involving not only intravasation of tumor cells from the primary tumor into circulation, but extravasation from circulation into the bone where they meet an environment that is generally suppressive of their growth. The bone microenvironment can inhibit the growth of disseminated tumor cells (DTC) by inducing dormancy of the DTC directly and later on following formation of a micrometastatic tumour mass by inhibiting metastatic processes including angiogenesis, bone remodeling and immunosuppressive cell functions. In this review we will highlight some of the mechanisms mediating DTC dormancy and the complex relationships which occur between tumor cells and bone resident cells in the bone metastatic microenvironment. These inter-cellular interactions may be important targets to consider for development of novel effective therapies for the prevention or treatment of bone metastases.

## 1. Bone Metastasis

Bone marrow is a primary lymphoid organ that fills the medullary cavity in bones. It is highly vascularized being intersected by arterioles and capillaries and surrounded by calcified bone. Bone marrow contains unique vasculature structures called sinusoids, which are lined by an endothelial cell layer and interconnected by a network of sinusoidal arteries [1]. The bone marrow space is subdivided into two components: the hematopoietic parenchyma and the vascular stroma [2]. The hematopoietic parenchyma is located in close proximity to both the endosteum and blood vessels and is generally where hematopoietic stem and progenitor cells reside, clarifying this region as the hematopoietic stem cell niche [3]. Quiescent hematopoietic stem cells reside near arterioles, as opposed to sinusoids, prior to their differentiation into cells of the blood [3], whereas the vascular stroma region is comprised mainly of blood vessels, adipocytes, osteoblasts, osteoclasts, immune cells and mesenchymal stem cells (MSC). MSC can give rise to adipocytes, osteoblasts and chondrocytes.

Bone is a favorable site for metastatic cancer growth and is the most common metastatic site in patients with breast cancer [4] and prostate cancer [5]. Bone metastases also arise in patients with lung cancer, multiple myeloma, neuroblastoma, renal cell carcinoma and thyroid cancer [6,7,8,9,10]. Disseminated tumor cells (DTC) in the bone preferentially locate to the hematopoietic stem cell niches of the bone [11,12], where they have been shown to compete for space with the normal cellular occupants of the niche, which they may do in part by promoting the terminal differentiation of stem or precursor cells [11].

Through analysis of patient bone scans, bone metastases have shown differential localization preferences throughout the skeleton depending on the primary tumor type, however some consistent patterns arise. For example, bone metastases arising from primary breast, prostate and lung cancers most often develop in the spine, ribs and pelvis [5,6,13,14]. Metastases in breast cancer patients also commonly arise in the sternum [15], but similar to lung cancers may also arise in the femur, skull, humerus, scapula and/or clavicle [6,13,15]. Interestingly, while it has been shown in lung cancer that there are no significant differences in bone metastatic ability and site preference between pathologic subtypes [6], it is well known that luminal breast cancer subtypes spread to the bone more frequently than triple negative or Her2+ breast cancers [16]. Bone metastatic lesions may also be described as either osteoblastic or osteolytic, depending on the pathological activity of osteoblasts and osteoclasts. Osteoblastic lesions are more common in prostate cancer and osteolytic lesions are more common in breast and lung cancer, however a combination of both types of lesions may be present in all [17,18,19]. Osteoblastic lesions are characterized by overactivity of osteoblasts leading to increased bone formation resulting in hardened or osteosclerotic bone tissue, often accompanied by underlying weak or poorly constructed bone [20,21]. Osteolytic lesions are characterized by overactivity of osteoclasts, leading to breakdown of the bone [22,23]. Both types of lesions are associated with bone pain, an increased risk of fracture and hypercalcemia [19,21,24,25,26,27,28]

Prior to formation of bone lesions, DTC are commonly detected in the bone of patients with breast cancer early on in disease progression without otherwise evidence of metastatic disease; this is a positive indicator of relapse and poor prognosis [29,30,31,32,33,34,35,36]. DTC have also been detected in 72% of newly diagnosed prostate cancer patients prior to radical prostatectomy and DTC detection was associated with recurrence in this tumor type [37]. Despite seeding the bone early during disease progression, DTC appear to lie dormant in the bone for extended periods of time (normally ~1–3 years for lung cancer [38,39] and >5–10 years for breast [36,40,41,42] and prostate cancers [37,43]) before micro and macrometastatic tumor growth occurs [44,45,46]. 

Upon extravasation of DTC to the bone marrow, the DTC will either undergo apoptosis, find a suitable niche for growth or enter a state of solitary cell dormancy [11,47,48,49]. DTC exist in a quiescent state within the hematopoietic stem cell niche, fluctuating between the G0 and G1 phase of the cell cycle [11,49,50,51,52]. In this state of solitary cell dormancy, the DTC are resistant to chemotherapy and can avoid immune detection [53,54,55,56]. Multiple cell intrinsic mechanisms may be employed by DTC to initiate or maintain a dormant state. One cellular change is in the ratio of p38 to ERK signaling: simultaneous increased p38 activity and decreased ERK activity is associated with DTC dormancy [51,57,58]. It has also been shown that cyclin D1 expression which mediates proliferation of tumor cells is a result of EGFR autophosphorylation leading to its complexing with PI3K and subsequent activation of ERK and AKT, and that under stress conditions such as growth factor deprivation, tumor cell dormancy is promoted as a result of decreased active AKT and cyclin D1 levels due to failed ERK-PI3K complex formation [59]. DTC survival in the hypoxic environment of the bone marrow may also be regulated by suppressed AKT activity as it was shown that cells which survived under long term hypoxic conditions did so by inducing a state of dormancy mediated by downregulation of AKT [60]. Bone resident cells play a role in enforcing the quiescence of the DTC. For example, osteoblasts have been shown to induce dormancy in prostate cancer tumor cell lines PC3 and DU145 through transforming growth factor (TGF)-β signaling [61]. MSC, which can differentiate into osteoblasts, have been shown to produce miRNA-containing exosomes or express soluble factors that alter tumor cell signaling and translation resulting in promotion of quiescence in tumor cells [62,63,64]. This period of microenvironment-induced DTC quiescence may eventually be overcome and the DTC will proliferate and form a micrometastasis. Micrometastases in the bone may also be inhibited from further growth through a process described by some as ‘tumor mass dormancy’ [65]. In order to overcome tumor mass dormancy, micrometastases create new or co-opt existing vasculature via the process of angiogenesis, induce bone remodeling resulting in formation of a tumor niche equipped to release matrix-bound growth factors, and continually evade the anti-tumor functions of the immune system. As a result of extensive tumor cell crosstalk with bone-resident cells, micrometastases may overcome tumor mass dormancy to form a highly bone-destructive and pain-inducing macrometastasis.

## 2. Current Therapy for Bone Metastases

Current treatment strategies for bone metastases are designed to attempt to induce tumor regression, inhibit tumor cell growth, or alleviate the effects of tumor in the bone that can lead to bone metastasis related complications, also known as skeletal related events (SRE). SRE can include fracture, bone pain, spinal cord compression and hypercalcemia [66,67]. Current bone metastatic therapies are largely palliative as opposed to curative, for reasons that remain elusive but may be due in part to the inability of therapeutic agents to achieve effective concentrations in the bone, the hypoxic nature of the bone microenvironment, and the use of treatments which only target rapidly proliferating cells, leaving slowly proliferating or dormant tumor cells unaffected. Commonly used therapeutic interventions for bone metastasis will be discussed briefly below and include chemotherapy, bone targeting agents, radiation therapy, and surgical interventions.

Chemotherapy is used to treat both primary and metastatic cancer, however, achieving positive clinical outcomes for bone metastases remains challenging. As the clinical benefits of chemotherapy are often assessed by their ability to affect primary tumor cell growth for approved use in patients, the organ-specific effect of chemotherapy, including the bone, is rarely analyzed. Anthracyclines as a chemotherapy, including doxorubicin, inhibit topoisomerases and are employed to treat patients with metastatic prostate cancer, breast cancer and lung cancer, among others [68,69]. Taxane derivatives, such as paclitaxel, cabazitaxel and docetaxel, are also routinely utilized to target these types of cancer with the mechanism of action involving inhibition of mitotic spindle formation [70]. Chemotherapies are routinely employed in combination to enhance clinical outcomes. Although the combination of doxorubicin chemotherapy and the bone-targeting agent (BTA) zoledronic acid resulted in decreased tumor burden in in vivo pre-clinical bone metastatic models [71,72], combination of chemotherapy with BTA in bone metastatic patients has generally had no benefit on overall survival, but does however prevent SREs [73]. Bone metastases remain largely unimpacted by most chemotherapies and the bone itself may be a difficult organ to target due to its complex and variable structure, including highly abundant cartilage adjacent to vasculature which could impede drug delivery [74]. In fact, it has been shown that in some cases <1% of delivered drug reaches the bone microenvironment [75]. Nonetheless, some studies have provided potential indications of the positive effect of some chemotherapies or the use of combination therapies as opposed to monotherapies on bone metastases. Among bone-only metastatic breast cancer patients, those with hormone receptor (HR)+ status receiving combination therapy (combination of anthracyclines, taxanes and endocrine therapy, with or without trastuzumab) had longer survival compared to those receiving monotherapy and the HER-2+ patients had longer progression-free survival when treated with trastuzumab compared to chemotherapy without trastuzumab; however this study is limited by the small patient cohort [76]. In patients with bone-only metastatic prostate cancer, chemotherapy combined with androgen blockade (docetaxel or abiraterone acetate) has been shown to be associated with increased median overall survival [77]. Further clinical analysis of treatment outcomes amongst bone-only metastatic patients or routine inclusion of bone lesion assessment in bone-metastatic patients before and after chemotherapy within clinical trials is warranted in order to analyze bone metastatic-specific effects of chemotherapy.

Current chemotherapies are generally also ineffective at targeting dormant DTC within the bone marrow [78,79,80]. Dormant tumor cells are more resistant to chemotherapy than actively proliferating tumor cells due to the mechanism of action employed by most chemotherapeutics being to target and inhibit cellular proliferation. One proposed mechanism by which dormancy confers chemotherapeutic resistance is that stromal cells can induce expression of TBK1, which has been shown in PC3 and C4-2B tumor cells to promote chemotherapeutic resistance via inhibition of mTOR [81]. Outside of the bone metastatic microenvironment, increased p38 activity in dormant human epidermoid carcinoma HEp3 cells has been shown to confer resistance to doxorubicin and etoposide chemotherapy regimens by increasing pro-survival signaling orchestrated by simultaneous activation of PERK and deactivation of Bax, the latter occurring as a consequence of BiP up-regulation [82]. Therefore, there may be multiple mechanisms by which dormancy confers resistance to chemotherapy, which demand further study in models of bone metastasis.

At the same time, chemotherapy can cause off-target effects on the bone resulting in skeletal degeneration as evidenced by various in vivo models indicating either increased osteoclast activity or decreased osteoblast activity in response to chemotherapy [83,84,85]. Thus, chemotherapy may be exacerbating the osteolysis already endemic to this pathology. In a murine model featuring administration of a combination therapy of cyclophosphamide, epirubicin and 5-fluorouracil, increased osteoclast presence in the metaphysis of the trabecular bone was observed [85]. Doxorubicin has also been shown to result in bone loss in patients with breast cancer [86], which has been further studied in in vivo preclinical models and indicates that doxorubicin may be responsible for inhibition of osteoblast differentiation and viability [87]. This chemotherapy-induced skeletal degeneration may further contribute to the occurrence of SREs or further prime the bone for metastatic growth upon DTC overcoming their microenvironment-induced dormancy.

Bone-resident cells can also provide a chemoprotective effect for nearby DTC or micrometastases. For example, it has been shown that coculture of the murine bone stromal cell line MS-5 offered chemoprotection against vincristine to lymphoma cells [88]. It has also been shown that MSC cocultured with breast tumor cells in 3D mineralized scaffolds had increased resistance to chemotherapies such as paclitaxel due in part to increased secretion of IL-6 which in turn upregulated expression of multi-drug resistance genes, such as MRP1 and ABCG2, in a STAT3-dependent manner [89]. Specific areas of the bone marrow adjacent to newly woven bone may be described as ‘resistant niches’ and have been shown to confer resistance to targeted therapies to tumor cells residing there [90]. In this study, the authors found that despite tumor regression induced by treatment with carbozantinib, tumors residing in areas adjacent to newly remodeled bone remained viable, and they further identified that resistance was due to enhanced tumor cell viability mediated by integrin activation of focal adhesion kinase (FAK). Although poorly understood, the mechanisms contributing to resistance to current therapies conferred by the bone microenvironment could provide effective new targets to consider for development of adjuvant chemotherapy in the management of bone metastatic disease. 

The most commonly used BTAs are the bisphosphonates, including zoledronic acid, clodronate and pamidronate. Bisphosphonates are administered primarily to patients with osteolytic lesions, as they induce apoptosis of osteoclasts and thereby decelerate weakening of the bone as a result of both the pathology itself and chemotherapy [91,92,93]. Bisphosphonates bind to calcium in the bone mineralized matrix where they impede function and decrease viability of osteoclasts via inhibition of the osteoclast mevalonate metabolic pathway [92,94]. Zoledronic acid has been found to be the most effective of the bisphosphonates at preventing SREs and thus is primarily utilized in current treatment regimens [95,96]. Another drug, denosumab, was FDA approved in 2010 for treating bone metastatic lesions as a monoclonal antibody that binds to receptor activator of nuclear factor kappa-Β ligand (RANKL). The binding of RANKL with its receptor (RANK) facilitates the differentiation of osteoclasts into their mature and active form at multiple points in the differentiation process [97], thereby providing another means to inhibit overall osteoclast induced bone lysis. Denosumab has been shown across a number of studies and tumor types to prevent the occurrence of SREs more effectively than zoledronic acid in patients with metastatic bone disease [98,99,100,101]. In addition, denosumab was shown in a phase 3 clinical trial (NCT00286091) among castration-resistant prostate cancer patients to increase bone-metastasis-free survival by 4.2 months, but did not lead to increased overall survival [102]. Therapeutic strategies including bisphosphonates and denosumab may offer some patients a temporary delay in the occurrence of SREs as a result of bone metastases or may even delay the onset of bone metastatic growth, however, do not necessarily prevent bone metastatic tumor growth, nor offer increased overall survival to patients.

Radiation can be performed to relieve pain caused by bone metastases [103,104]. Three-dimensional conformal radiation therapy (3-DCRT) is most often employed, however stereotactic body radiation (SBRT) is also utilized, particularly for patients with fewer metastatic lesions as it offers more precise targeting [104]. The α-emitter radiotherapy, radium-223 dichloride, was shown to increase overall survival by a median of 2.8 months compared to placebo in metastatic prostate cancer patients not able to receive docetaxel [105], and showed benefits associated with pain relief [106] and in delaying time to first SRE [107]. Strontium-89 chloride is a routinely utilized β-emitter, which also alleviates pain caused by bone metastases and can be used as an alternative when patient resistance to α-emitters occurs [108].

Use of surgery to treat bone metastases is sometimes employed but mainly as a palliative measure with the goal of reducing pain, improving mobility or stabilizing a fracture [109]. Due to the lack of proper bone formation and healing endemic to this pathology, more advanced interventions including use of cement and plates in surgical procedures are routine in these cases [109].

Current standard of care treatment may offer some benefit in terms of prolonging metastasis free intervals, progression free interval or interval to first SRE, however, to date they have shown no significant effect on patient overall survival. It is clear we urgently need to find effective treatments to not only enhance overall survival in bone metastatic patients, but to prevent their development or progression in the first case. To do this, we must gain a clearer understanding of the process of bone metastasis development and the complex inter-cellular communications that occur to facilitate bone metastasis progression. Bone marrow is a rich reservoir of cell types that may influence tumor growth and include immune cells, bone-forming osteoblasts, bone-degrading osteoclasts, endothelial cells and adipocytes. Below we will highlight some of the tumor cell - bone resident cell communication factors and mechanisms that regulate bone metastasis growth which could become future therapeutic targets.

## 3. Inter-Cellular Regulation of Bone Metastasis Growth

### 3.1. Role of Bone Remodeling in Bone Metastatic Tumor Growth

Bone remodeling is a normal physiological process that is dependent on the opposing abilities of osteoclasts to degrade bone and osteoblasts to promote formation and mineralization of new bone [110,111,112,113]. Many years ago, the ‘vicious cycle’ of bone metastasis was postulated by Mundy [114]. In radiologically osteolytic breast cancer, it was hypothesized that factors released during bone remodeling stimulate the production of osteoclastogenic factors by tumor cells, such as parathyroid hormone-related peptide (PTHrP), which in turn stimulate osteoclasts to degrade bone. This bone degradation causes further release of additional bone matrix-bound growth factors, such as TGF-β, which promote tumor growth. As tumors grow, they incite further osteoclast-mediated bone degradation to perpetuate the process. Although bone degradation was initially thought to be limited to those tumor types that promote osteolytic lesions, the presence of elevated levels of bone resorptive markers in prostate cancer patients and their association with outcome suggests that even tumors classified radiographically as osteoblastic bone metastases may also rely on osteolysis for their progression [115,116]. The overexpression of PTHrP is common in a number of different tumor types which show predominant metastasis to the bone and its expression can be upregulated by numerous growth factors including TGF-β, vascular endothelial growth factor (VEGF) and platelet derived growth factor (PDGF) [117]. PTHrP has been shown to have increased expression in bone metastasis as compared to counterpart primary tumors [118] and its forced overexpression in dormant MCF7 cells conferred more aggressive metastatic growth [119] which was associated with decreased tumor cell expression of genes associated with dormancy in a PTHR1/cyclic AMP-independent manner [120]. The role of PTHrP as a critical mediator of bone formation has been demonstrated in a number of transgenic and knock out mouse model systems [121,122,123]. Osteoblasts are a predominant source of PTHrP in bone, however exogenous tumor-derived PTHrP can also stimulate osteoblast production of RANKL which in turn promotes osteoclastogenesis and osteoclast activation leading to increased bone turnover [124,125]. PTHrP production by tumor cells has also been shown to be associated with an autocrine stimulation of VEGF production by PTHrP receptor expressing tumor cells [126], resulting in alterations in osteoblastogenesis and angiogenesis in favor of tumor growth, which will be discussed in more detail below.

As part of normal bone remodeling, osteoblasts and osteoclasts regulate each other’s function. This homeostasis is interrupted by the presence of DTC, which can modulate the function of osteoclasts and osteoblasts directly and in turn, osteoclasts and osteoblasts may also directly regulate tumor cell phenotypes. One mechanism altered in the bone marrow microenvironment upon DTC introduction and metastatic development is the Notch signaling pathway, which plays a role in osteoblast interactions with tumor cells leading to enhanced osteoclastogenesis and osteoclast activity. TGF-β1 expression by osteoblasts promotes DTC growth and expression of Notch3 and its ligand Jagged1, resulting in downstream phosphorylation of Smad3 and Smad2 in DTC. Notch3 expression by tumor cells has been suggested to play a role in formation of osteolytic lesions, as its suppression has been shown to reduce osteolysis of the bone in an in vivo bone metastasis model [127]. Inversely, osteoblast-osteoclast interactions as a result of tumor cell Jagged1 expression promote tumor growth [128,129]. Jagged1 has the ability within this context to induce osteoblast secretion of IL-6 which in turn promotes differentiation of osteoclasts and subsequently, their breakdown of bone, forming or accelerating formation of osteolytic lesions [128]. Zheng et al. [129] have since developed a Jagged1-targeting human monoclonal antibody, clone 15D11. 15D11 showed high affinity for mouse Jagged1, restricted osteoclast differentiation in vitro on recombinant Jagged1-coated plates and decreased mRNA expression and production of IL-6 in cultures of osteoblasts [129]. In in vivo models of bone metastasis involving injection of a highly metastatic MDA-MB-231 derivative cell line, administration of 15D11 was shown to inhibit osteoclast activity leading to a 5-fold decrease in bone metastatic burden accompanied by no significant increase in bone density, the latter of which could have otherwise possibly occurred due to excessive loss of osteoclast function. When administered in mice with already established bone metastases, the decrease in metastatic burden in mice treated with 15D11 was not significant, however was significant when combined with a denosumab equivalent, OPG-Fc, which itself alone also did not lead to a significant decrease in bone metastatic progression. In addition, 15D11 has shown no off-target effects based on measures of liver, GI tract and hematologic toxicity suggesting its use and targeting Notch pathway dependent breakdown of bone may have some future clinical therapeutic utility.

A variety of tumor cell surface receptors and membrane-associated proteins have been assigned osteogenic-stimulatory roles. In a subpopulation of MDA-MB-231 breast tumor cells that showed lack of ability to form bone metastases in vivo, authors could show that the ability to form progressive bone metastatic lesions was accompanied by increased expression of VCAM-1 by tumor cells [23]. They further showed that forced overexpression of VCAM-1 in non-bone metastatic tumor cells conferred the ability to form progressive bone metastases; this was attributed to VCAM-1 interaction with its receptor integrin α4β1 on osteoclasts resulting in enhancement of osteoclast activity [23]. Expression of another integrin, αvβ3 in tumor cells has also been implicated in development of bone metastases [130]. Overexpression of integrin αvβ3 in tumor cells was shown to increase bone metastatic burden and osteolysis, which could be inhibited by delivery of PSK1404, an integrin αvβ3 antagonist, in an in vivo bone metastasis model. Although the mechanism by which integrin αvβ3 regulated these processes was not clearly established by the authors, it has been shown that osteoblasts secrete osteopontin, a ligand of integrin αvβ3, and release of osteopontin into the extracellular matrix provides survival cues for dormant tumor cells. Osteopontin-integrin αvβ3 interactions have also been shown to promote critical mediators of tumor cell proliferation and survival including activation of Src, ERK and PI3K/AKT/mTOR pathways [125]. Integrin αvβ3 is expressed by osteoclasts as well, and osteopontin has been shown to promote osteoclastogenesis and promote bone resorptive activities of osteoclasts by enhancing podosome formation and secretion of proteases [131,132]. Osteopontin has also been shown to promote expression of VEGF by tumor cells [133]. Thus, integrin α4β1 and αvβ3 with their respective ligands VCAM-1 and osteopontin offer potential targets for therapeutic development within the osteoblast-osteoclast-tumor interface.

VEGF has been shown to be a major factor contributing to osteoblast and osteoclast interactions within the bone marrow. Although VEGF is well characterized for its role as an angiogenic growth factor, VEGF can also promote differentiation of MSC into osteoblasts, as opposed to adipocytes [134]. VEGF present in C4-2B metastatic prostate cancer tumor cell conditioned media enhanced osteoblast mineralization, which was decreased when treated with anti-VEGF antibody or when recombinant noggin (an inhibitor of BMP) was present, indicating that tumor cell-derived VEGF and BMPs promote osteoblast activity [135]. These findings have been further recapitulated in an in vivo bone metastasis model which showed that although VEGF alone is capable of inducing osteoblast differentiation, it alone is not sufficient to induce osteoblast mineralization and as such additional tumor-produced factors yet to be identified are required for tumor-induced osteoblast mineralization [136]. As well, a VEGF homolog, placental growth factor (PlGF), is expressed and secreted by both bone-resident cells and tumor cells [137,138,139]. PlGF levels were found to increase over time in both tumor cell and bone stromal cell fractions following injection of MDA-MB-231 cells into mice in a bone metastatic model [140]. The neutralization of P1GF by administration of the antibody 5D11D4 led to decreased formation and size of osteolytic metastases in murine models [140,141], believed to be in part due to reduced levels of the osteoclast activators, RANKL and C-telopeptide. Furthermore, P1GF-null bone marrow MSCs showed impaired ability to express RANKL as they exhibited a 6-fold lower level of RANKL when cultured with MDA-MB-231 cells compared to PlGF-expression MSC [140]. These pre-clinical studies show potential therapeutic efficacy for drugs inhibiting VEGF or P1GF growth factors or related pathway members in the context of tumor-osteoblast-osteoclast intercellular signaling. By recognizing the many pathways regulating bone remodeling, therapy to prevent or halt the bone-destructive mechanisms instigated by tumor cells or to counteract these mechanisms would not only prevent SRE occurrence but may also interfere with growth of bone metastatic tumor masses.

### 3.2. Angiogenesis in Bone Metastasis

Angiogenesis is the process of new blood vessel formation. Micrometastases arising from re-animated DTC in the bone require neovascularization to grow beyond a cubic millimeter in size [142,143,144]. It is also well documented that tumors with higher expression levels of angiogenic factors are more metastatic [145,146,147,148,149], presumably in part due to their intrinsic ability to promote neovascularization in metastatic sites. The vasculature in bone contains many fenestrated sinusoids which can facilitate hematopoietic cell trafficking [150], and the endothelium appears to constitutively express many cell surface adhesion proteins that may facilitate extravasation of tumor cells into the bone via increased binding. The vasculature has also been shown to express the angiogenic factor PlGF [151,152], which can enhance tumor cell adhesion to fibronectin during early stages of metastasis [153]. In addition to the vasculature facilitating tumor cell metastasis to the bone, endothelial cells of the vascular niche have been shown to express high levels of thrombospondin-1 (TSP1) which can inhibit tumor cell proliferation and therefore the vascular niche may also directly promote DTC dormancy in a TSP1-dependent manner [153]. The mechanisms by which the tumor-endothelial cell interactions are altered to overcome DTC dormancy and promote growth beyond micrometastasis remain unclear. However, it is known that endothelial cells will downregulate TSP1 and upregulate factors such as TGF-β, periostin and fibronectin, which may promote tumor cell survival and facilitate additional production of tumor-derived angiogenic factors. Thus, there is substantial evidence in support of promotion of both tumor growth and dormancy by existing and new vasculature.

DTC themselves also may express angiogenic factors to aid in the process of angiogenesis. Multiple angiogenic factors are over-expressed in highly metastatic prostate cancer cell lines, including VEGF, ICAM-1, IL-8 and TGF-β2 and expression of factors such as VEGF and ICAM-1 may be dependent on MMP-9 expression [146]. DTC may also regulate angiogenesis by decreasing expression of angiostatic factors, as it has been shown that highly angiogenic derivatives of breast adenocarcinoma, osteosarcoma and glioblastoma cell lines had lower expression of TSP1 [154].

As a highly hypoxic tissue, the bone marrow can facilitate angiogenesis under hypoxic conditions by mechanisms similar to that shown in primary tumors. Numerous angiogenic factors are upregulated by hypoxia in the bone metastatic environment, including VEGF, PlGF and CXCL8, and it has been shown that blocking HIF activity inhibited the progression of breast cancer induced bone metastases in xenograft models [155]. Bone metastatic tissues of prostate cancer patients commonly exhibit increased CREB phosphorylation resulting in increased VEGF expression, which is dependent at least in part on HIF [156]. In addition to tumor cell promotion of angiogenesis in the bone, osteoblasts and osteoclasts have also been shown to participate in angiogenesis. Both express isoforms of VEGF-A, a major pro-angiogenic factor [157,158,159,160,161], and VEGFR1 and VEGFR2 [162]. Release of matrix metalloproteinases by osteoclasts, including MMP-9, has been shown to enhance angiogenesis and thereby aid in growth of PC3 prostate cancer cells in a murine model [163]. Conditioned media from MMP-9 null osteoclasts had decreased VEGF-A expression and decreased ability to promote angiogenesis in an aorta sprouting assay [163]. Osteoclast release of osteopontin has been shown to enhance angiogenesis in a model of multiple myeloma [164]. Addition of isoproterenol, an activator of the beta 2-adrenergic receptor (b2AR), to osteoblasts resulted in increased secretion of VEGF-A, which not surprisingly, increased angiogenesis in murine bone metatarsal explants [165], suggesting a potential for the use of beta-blockers in prevention of angiogenesis in the bone metastatic environment. Bone resident cells may also inhibit angiogenesis in the bone as osteoblasts and osteoclasts also express pigment epithelium derived factor (PEDF) [162], a major negative regulator of VEGF-A. In neo-vascularization in non-pathological settings, bone marrow MSC isolated from patients have increased levels of PEDF compared to VEGF, however the ratio of VEGF/PEDF is enhanced under hypoxic conditions, suggesting that bone marrow MSC may normally have an anti-angiogenic role but in contrast may promote angiogenesis under hypoxic conditions [166]. In addition to direct effects on endothelial cells, PEDF may affect other bone resident cell types. Treatment with recombinant PEDF was shown to upregulate expression of osteoprotegerin and thereby reduce differentiation of precursor cells into mature osteoclasts which in turn would impair breakdown of bone as a consequence [167]. PEDF-derived peptides have also shown efficacy in vitro in reducing expression of VEGF and promoting differentiation of MSC to osteoblasts resulting in inhibition of tumor growth in an in vivo sarcoma model [168]. Recombinant PEDF was also shown to decrease the skin wound capillary density in mice [169]. Taken together, these studies indicate a role for PEDF as an inhibitor of vascularization in the bone metastatic environment.

Due to the pivotal role of angiogenesis for micrometastatic growth, inhibition of angiogenic factors is a therapeutic strategy of interest. An oral VEGF-inhibitor, cediranib, was shown to decrease bone metastatic tumour burden in murine models [170]. Initially, cediranib was shown to have some anti-tumour ability in extensively pretreated patients with metastatic castration-resistant prostate cancer in a phase II clinical trial [171], however in a larger, more recent phase II clinical trial no benefit to progression-free survival or overall survival was observed [172]. Additional ongoing clinical trials are analyzing cediranib with olaparib in metastatic castration-resistant prostate cancer (NCT02893917) and metastatic BRCA mutated breast cancer (NCT04090567), however results have not as yet been reported. Although angiogenesis within the bone metastatic environment is a coordinated process involving tumor cells and bone-resident cells and is highly dependent on the specific gene expression profile of the tumor, given the requirement for angiogenesis for the tumor mass to grow, targeting of specific angiogenic factors or intercellular interactions within the microenvironment remain of interest in preventing the further development and severity of metastatic bone lesions. 

### 3.3. Immune Cell Function in Bone Metastasis

The bone marrow is a predominant site for immune cells and an exceedingly important location harboring a number of immune precursor cell populations. Given the vast immune repertoire in the bone marrow, it is not surprising that immune-mediated anti-tumor effects are a predominant mechanism controlling tumor-cell dormancy in the bone. Although not completely well characterized, it has been shown generally that CD8+ T-cells, NK cells and macrophages can mediate anti-tumor responses in the bone. However, immune cell types such as M2 macrophages, myeloid derived suppressor cells (MDSC) and T-regulatory cells (Tregs) have been shown to also promote bone metastatic tumor growth. To avoid the immune-mediated suppression by T-cells and NK cells, DTC in the bone may intrinsically downregulate MHC class I molecules to avoid detection [173]. DTC may also upregulate expression of PD-L1 to regulate the function of T-cells and NK cells through checkpoint inhibition [174]. Tumor cell upregulation of anti-apoptotic factors such as Bcl-2 has been demonstrated to be higher in bone metastases compared to other soft tissue metastases in prostate cancer patients [175] and given it has been suggested that inhibition of Bcl-2 can sensitize tumor cells to T-cell mediated killing [176], upregulation of Bcl-2 family proteins may be another mechanism by which tumors can overcome immune-mediated dormancy and suppression of growth in the bone. Tumors may also influence immune cell phenotypes and functions to support their growth in bone. We will highlight some of these mechanisms in specific immune cell phenotypes below.

#### 3.3.1. T-Cells

In the bone, B-cells and T-cells represent approximately 10–20% of the monocyte cell population. In addition to the intrinsic mechanisms that may allow DTC to escape immune detection as described above, tumor cells may also actively render cytotoxic T-cells less effective via secretion of TGF-β, a factor that is also secreted by or released from mineralized bone by activated osteoclasts. Thus, tumors can influence T-cell phenotype either directly through TGF-β or indirectly by inducing osteoclastogenesis. It has been shown that TGF-β induces downregulation of key factors involved in cytotoxic T cell-mediated cytotoxicity, including perforin, granzyme A, granzyme B, Fas ligand, and interferon gamma [177]. TGF-β can also induce the CD4+/CD25+/FOXO3P+ Treg cell phenotype in CD4+ T-cells to induce immunosuppression [178]. Tregs may also promote differentiation of monocytes into M2 macrophages [179] which themselves can be immunosuppressive and promote tumor growth. Tregs have been shown to be important in cancer and often correlate with metastasis and poor patient prognosis [180,181,182]. Treg numbers are increased in prostate cancer patients with bone metastasis [183], and this may be due in part to active recruitment and retention of Tregs in the bone marrow by CXCR4/CXCL12-mediated signaling [184], a pathway that is often also upregulated in DTC [185,186,187]. In addition to their function in suppressing the activity of T-cell cytotoxicity, which in part may also be due to secretion of TGF-β [188,189,190,191], Tregs also secrete high levels of RANKL which facilitate the tumor-induced process of osteoclast activation, and promotes a favorable pre-metastatic niche in bone by inducing osteolysis and release of bone matrix bound growth factors prior to tumor cell arrival [192,193]. Moreover, Tregs facilitate bone metastasis growth by influencing angiogenesis. It has been shown that Tregs can express high levels of VEGF [194], and that there is a direct association with Treg and vessel number in tumors [195,196]. Taken together, these findings suggest that Tregs may play a crucial role in mediating immune suppression, osteolysis and angiogenesis to facilitate overcoming tumor bone microenvironment induced dormancy.

#### 3.3.2. Macrophages

It is well established that macrophages can be polarized into different cell phenotypes including the M1 macrophages, which are generally anti-tumorigenic, and the M2 macrophages with are generally immunosuppressive and thus more pro-tumorigenic. The intermediate phenotype M2-like macrophages share some features specific to each polarized type. It has been shown that tumors may promote macrophage differentiation into a M2-like phenotype through secretion of CCL2 and IL-6 [197]. One of the factors secreted by these M2-like macrophages which may be exceedingly important for bone metastatic growth is TGF-β [198], which as discussed above can affect tumor cell growth, osteolysis, immune suppression and angiogenesis in the bone. M2 macrophages can also promote angiogenesis via secretion of VEGF, EGF and IL-8 angiogenic factors [199,200,201,202]. M2 macrophages also produce CCL22 which is a potent chemoattractant for Tregs [203]. In addition to these macrophage types, there appears to be a more specialized bone macrophage termed osteomac [204]. Osteomacs appear to reside along the endosteal surface of the bone, the same predominant location of bone metastatic tumor growth, and are increased at areas of remodeling bone [205,206,207] where they are also thought to regulate osteoblastogenesis [208,209,210,211,212]. In the bone, osteomacs also sense apoptotic cells and may engulf them or secrete a number of immune cell chemoattractants such as CCL2 and M-CSF [213], or factors that induce osteoblastogenesis such as TGF-β [213,214,215]. Along with this, an important role for macrophages in bone metastasis progression has also been demonstrated. For example, macrophage depletion prior to intratibial tumor cell inoculation resulted in decreased tumor growth in the bone [216], and decreased tumor-induced bone formation in prostate cancer models [217]. Given the ability of macrophages to regulate processes that are critical for bone metastasis progression including immune cell modulation, angiogenesis and bone remodeling, they may be an important therapeutic consideration in inhibiting bone metastasis progression and maintaining tumor mass dormancy.

#### 3.3.3. MDSC

MDSC are primarily non-macrophage immature myeloid cells that arise in the bone marrow and have demonstrated potent immunosuppressive functions [218]. MDSC express CD11b and CD33, and depending on the subtype will also express CD14, CD15 or CD66b surface markers. In cancer, tumor-derived expression of IL-1, IL-6 and TNFα have been shown to promote accumulation and immunosuppressive abilities of MDSC [219,220,221,222,223]. It has also been shown that tumor derived prostaglandin E2 (PGE_2_) production stimulates differentiation of MDSC from precursor stem cells in the bone [224,225]. MDSC have been shown to suppress T-cell activation [226,227,228,229,230], polarize macrophages to a pro-tumorigenic phenotype [231,232] and recruit Tregs [233,234]. In addition to their well described role in immunosuppression, MDSC have also been shown to confer stem-like phenotypes to tumor cells through various mechanisms including MDSC secretion of IL-6, activation of STAT3 or induction of miRNA expression in tumor cells [235,236,237]. MDSC may also promote angiogenesis via their ability to secrete VEGF [238]. Importantly, as MDSC arise from the same lineage as osteoclasts, it has been shown that they can be induced to differentiate into osteoclasts, but only in the presence of tumor bone metastases, as MDSC isolated from the bone marrow of tumor-free mice or from those harboring lung metastases instead were unable to be differentiated into osteoclasts in vitro [239]. Although the exact mechanism remains unclear, the authors showed that upregulation of nitric oxide production occurred in MDSC isolated from mice bearing bone metastatic tumors, and its blockade prevented differentiation into osteoclasts. Although not in the context of bone metastases, use of low-dose DNA methyltransferase and histone deacetylase inhibitors as epigenetic inhibitors in an adjuvant setting, resulted in prolongation of survival in part by inhibiting the CCR2 and CXCR2-dependent trafficking of MDSCs to pre-metastatic niches in the lung and promoting their further differentiation in macrophages [240]. These findings, and the ability of MDSC to affect multiple processes and cell types involved in bone metastasis progression suggest they may be an important therapeutic target to consider.

## 4. Rethinking Therapy for Bone Metastases

As described, bone metastasis formation and growth consist of multiple distinct stages which involve multiple cell types and signaling pathways. Thus, although the bone metastatic process provides a plethora of potential targets, the efficacy of specific treatments may potentially differ between patients based on which metastatic stage they are in at the time of treatment. Upon extravasation to the bone marrow, DTC may undergo a period of solitary cell dormancy which is reinforced by bone-resident cells and can last for many years as discussed in previous sections. Due to the extended period of bone microenvironment-induced dormancy of DTC, novel therapeutic targets which can aim to induce or elongate this period of solitary cell dormancy may be beneficial to prolong patient overall survival. In addition to attempting to maintain DTC dormancy, approaches that target factors which play roles in multiple pathways that control bone metastasis growth simultaneously may also be effective treatments to inhibit bone metastasis growth at early stages of its progression. Although numerous candidates to target these processes have been identified, we will discuss a few which may have pleiotropic activities which are highlighted in Figure 1 and summarized further in Table 1.

### 4.1. NR2F1 as an Inducer of Dormancy

With regard to inducing tumor cell dormancy, Sosa et al. have shown that the nuclear receptor NR2F1 which participates in transcriptional regulation [300] is upregulated in DTC in 43% of prostate cancer patients with dormant disease compared to 10% of DTCs in patients with advanced metastatic disease [293]. The authors further showed that a demethylating agent in combination with retinoic acid was capable of inducing NR2F1-mediated dormancy in multiple human cancer cell lines seeded within the murine bone marrow [293]. A recent study has also correlated high NR2F1 expression in DTC with increased disease-free intervals in breast cancer patients [301]. These findings present NR2F1 activation as a novel potentially viable therapeutic target for inducing or sustaining dormancy of DTC within the bone marrow and warrant further investigation.

### 4.2. The Gas6-Axl-TGF-β Axis

Another strategy to induce dormancy in tumor cells may be via targeting the ability of bone resident cells such as osteoblasts to promote or maintain tumor cell dormancy. Axl expression in prostate cancer cell lines has been shown to be required for regulation of TGF-β signaling and for TGF-β2-mediated induction of tumor cell dormancy induced by osteoblasts following release of Gas6 [61]. This study suggests that targeting of the Axl-TGF-β-Gas6 axis may be an effective strategy for inducing dormancy in tumor cells. Interestingly, administration of propranolol, a beta-blocker currently approved for use in treating high blood pressure, to cultures of MC3T3-E1 osteoblasts resulted in their increased expression of Gas6 [255]. As it has been previously suggested that release of norepinephrine by the sympathetic nervous system can induce reactivation of dormant DTC via suppression of Gas6 [302], and that propranolol remained effective in upregulating Gas6 expression even in the presence of norepinephrine [255], these findings suggest that treatment of patients at high risk of developing bone metastasis could benefit from adjuvant use of propranolol. However, the efficacy of this drug in maintaining tumor cell dormancy in the bone in vivo remains to be demonstrated. Additional approaches to activate Axl in vivo have been shown by use of administration of recombinant Gas6 [256] or agonistic antibodies to Axl [257], however the efficacy of these approaches in promoting DTC dormancy in bone metastasis models has not as yet been tested. Should these approaches be pursued, it will be critical to assess the effects of these approaches in immune competent model systems as although Axl agonism may promote NK cell development and enhanced function [257] which could be beneficial for anti-tumor immune responses, Gas6-Axl activity has also been shown to promote the suppressive functions of Tregs in mice [303] which as discussed could enhance tumor growth in the bone.

Although the tumor cell expression of TGF-β mediated by the Gas6-Axl pathway can sustain quiescence, TGF-β also has well established functions in promoting tumor cell growth in addition to modulating bone remodeling, angiogenesis and immune cell function. This makes TGF-β a critical player in the bone metastatic process and thus a potentially valuable therapeutic target. However, given its possible roles in both suppression and promotion of DTC growth, its targeting would have to be considered carefully in a context-dependent manner. While the factors that alter TGF-β function from a tumor suppressor to a tumor promoter remain unclear, some mechanisms have been suggested. It has been shown that MED12 suppresses the response of tumor to TGF-β in part by its ability to regulate the levels of surface expression of TGFβRII [304]. In MED12 expressing cells, TGFβRII is bound by MED12 and remains unglycosylated and thus immature thereby preventing its expression at the surface. Suppression of MED12 or forced overexpression of TGFβRII resulted in TGF-β induced ERK activation, EMT phenotype and enhanced resistance to drugs including cisplatin and the targeted agents crizotinib and gefitinib. A parallel mechanism has been shown whereby the cytoplasmic protein PMEPA1 can act as a negative feedback loop regulator following its induction by TGF-β signaling in a SMAD3 dependent way [305]. Following upregulation of PMEPA1, it binds to R-SMAD proteins and sequesters them in the cytoplasm to inhibit downstream TGFβR signaling. The authors could further show that PMEPA1 depletion resulted in increased prostate cancer bone metastasis growth. Although these authors did not examine the effect of R-SMAD modulation in this context, others have shown that knockdown of SMAD3 resulted in impaired growth of breast cancer which demonstrated a prolonged ‘lag phase’ in their in vivo bone metastatic growth, while knockdown of its counterpart SMAD2 resulted in promotion of bone metastatic growth [258]. Although these authors did not assess DTC dormancy in their model, they could attribute the growth differences observed were due in part to differential regulation of angiogenic factors, whereby activity of SMAD3 seemed to promote VEGF production in tumor cells while SMAD2 inhibited it. These findings suggest that interfering with SMAD3 activity may be important in order to inhibit the pro-tumorigenic properties of TGF-β while potentially leaving its tumor-suppressor properties intact. Cell-penetrating peptides that specifically target SMAD3 have been generated by creating fusion proteins of the SMAD3 binding domains of SNX9 (sorting nexin 9) to the HIV TAT protein [264]. SNX9 is involved in protein trafficking and mediates the nuclear import of SMAD3 but not SMAD2 [265]. Although not tested in cancer metastasis models, the authors could show that the inhibitor peptide prevented nuclear accumulation of SMAD3 and inhibited growth of cells in soft agar assays. The SMAD3 inhibitory peptide also prevented expression of the SMAD3 targets plasminogen activator 1 (PAI-1) and CTGF (connective tissue growth factor). This has implications for bone metastases treatment as PAI-1 is a modulator of angiogenesis [260], and CTGF is induced by PTHrP stimulation and is also able to promote angiogenesis [261]. Moreover, inhibition of CTGF with neutralizing antibodies resulted in reduced growth of breast cancer bone metastasis in xenograft models [266]. Thus, use of SMAD3 specific peptide antagonists may have some therapeutic value in bone metastasis treatment.

Blocking TGFβR signaling has also been shown to inhibit skeletal metastasis growth. This was first evidenced in breast tumor cells in vitro: expression of a dominant negative TGFβRII lacking a cytoplasmic domain could eliminate TGF-β induced signaling that reduced bone metastasis growth in vivo, which was further attributed to reduced PTHrP expression by tumor cells [259]. Similar results were observed with the use of a pan-TGF-β neutralizing antibody ID11. Initiation of ID11 treatment shortly after tumor cell implantation was shown to delay progression of breast tumor bone metastases, as was treatment with the TGFβR1 inhibitor LY2109761 [262]. Interestingly, the authors showed that treatment with the receptor inhibitor effectively blocked the SMAD3 targets PAI-1 and CTGF, however antibody-mediated neutralization of TGF-β did not. LY210761 receptor inhibition was also shown to prevent tumor growth, osteoclast activity and associated bone loss in a prostate cancer bone metastasis model [306]. Taken together, these findings suggest that perhaps targeting the receptor or SMAD3 directly may be more effective approaches due to their ability to subsequently inhibit other pathways involved in bone metastasis growth and progression.

Unfortunately, despite these early promising studies, use of small molecule inhibitors in clinical trials in patients have shown dose-limiting toxicities, including serious cardiotoxicity in preclinical testing with the TGFβR1 inhibitor galunisertib (LY2157299) [263]. Although dosing in clinical trials was timed to avoid cardiotoxicities [307], limited efficacy was observed as a monotherapy or in combination with other chemotherapies [308,309]. However, in hepatocellular carcinoma, use of galunisertib in combination with sorafenib did prolong overall survival [310], suggesting its use in combination in bone metastasis patients may have some efficacy, although this remains to be tested. In addition to small molecule inhibitors, humanized antibodies targeting the ligands or receptors of the TGF-β pathway have also been tested. LY3022859, an antibody agent targeting TGFβRII resulted in significant toxicity including uncontrolled cytokine release syndromes in some patients [311]. Use of fresolimumab, an antibody that recognizes all three TGF-β isoforms, has shown better tolerability and some efficacy in patients [267,268]. In metastatic breast cancer patients, it was shown to prolong overall survival at the highest dose tested, and additionally patients treated at this dose level showed increased levels of CD8+ memory T cells [269]. These results hold some promise for its use in bone metastatic patients, however given the pleotropic nature of TGF-β activity careful design and dosing schedules are likely required to achieve the appropriate therapeutic window without adversely impacting patient quality of life. As discussed above, targeting factors downstream of TGF-β signaling may also be therapeutically useful, however to date, clinical studies assessing molecules such as SMAD3 antagonists have yet to be reported.

### 4.3. CXCL12/CXCR4 Signaling

Another therapeutic strategy to maintain bone marrow microenvironment-induced dormancy of DTC may be to target the CXCL12/CXCR4 axis. In addition to what has been discussed above, it has also been suggested that delivery of CXCL12 to chemotherapy-induced dormant neuroblastoma cells led to down-regulation of genes associated with cell quiescence [294], and dormant breast tumor cells in lung metastatic sites had reduced CXCR4 expression [295], reinforcing the role CXCL12/CXCR4 signaling may play in reanimation of dormant tumor cells in metastatic sites.

Lim et al. [49] showed that MDA-MB-231 and T47D breast cancer cell proliferation was blocked with cell cycle arrest in G0 by gap junction transport of the CXCL12-targeting miRNAs, miR-127, -222 and -223, upon coculture of healthy donor bone marrow stromal cells with tumor cells. This study potentiates delivery of CXCL12-targeting miRNA as a strategy for enhancing and maintaining dormancy of DTC prior to overcoming suppression by bone resident cells. However, pin-pointing the most effective timing for such an approach may be difficult as patients present with DTCs in the bone marrow early on in primary disease progression. Thus, the ability of these miRNAs to prevent growth or induce dormancy of micrometastases would be important to study in both an adjuvant and therapeutic setting. Inhibition of CXCR4 activation using Plerixafor, an antagonist that blocks CXCL12 binding to CXCR4 was shown to inhibit early bone metastasis growth in a prostate cancer model which was associated with reduced numbers of Ki67 positive tumor cells, suggesting possible induction of tumor cell quiescence, although markers of this were not directly assessed in this study [296]. Interestingly, Plerixafor had no impact on Ki67 levels nor any significant effect in reducing tumor burden in the bone when treatment was delayed until a time when tumors were established [296]. These findings suggest that CXCR4/CXCL2 targeting may only be effective early on in bone metastatic progression and may be inducing DTC dormancy, thus the timing of treatment for CXCR4/CXCL2 inhibition must be carefully examined to determine the most efficacious treatment strategy. It has also been shown that use of the CXCR4 inhibitor AMD3465 was effective in inhibiting bone metastasis growth in a breast cancer model [297]. Although the authors did not examine the effects of AMD3465 directly on tumor cell dormancy, they did observe that treatment with AMD3465 resulted in decreased numbers of intra-tumoral MDSC and Tregs and that systemic depletion of CD8+ and CD4+ T-cells allowed for tumor growth in the presence of AMD3465; these findings support the author’s contention that the anti-tumor drug efficacy was in part mediated by alleviating immunosuppression and promoting anti-tumor immunity in the tumor microenvironment. Inhibition of prostate cancer bone metastases was also observed following treatment with neutralizing antibodies to CXCR4 or delivery of a CXCR4 peptide antagonist, TC14012 in vivo [298]. Interestingly, although not in the context of bone metastasis, treatment with an oncolytic vaccinia virus targeting CXCR4 resulted in destruction of tumor vasculature concomitant with decreased levels of CXCL12 and VEGF within the 4T1 tumor mass, which overall resulted in increased tumor-free survival and decreased formation of metastases in a murine model [299]. Taken together, these findings suggest that targeting the CXCL12/CXCR4 axis may simultaneously affect DTC dormancy, angiogenesis and immunosuppression rendering it a worthy target for further investigation.

Clinical trials targeting CXCR4 signaling have recently been initiated and are still in early stages. A humanized monoclonal antibody to CXCR4, ulocuplumab (BMS-936564), has been tested in early phase trials and was well tolerated with some demonstrated clinically efficacy in multiple myeloma patients [312]. Use of the CXCR4 inhibitor plerixafor was shown to be well tolerated in multiple myeloma patients [313] and in metastatic colorectal and pancreatic cancer patients where it resulted in a tumor cell gene signature suggestive of response to immune-checkpoint mediated therapies [314]. Another peptide antagonist to CXCR4, BL-8040 (Motixafortide), has recently been tested in pancreatic cancer as a means of mobilizing bone marrow immune cells to enhance efficacy of the immune checkpoint inhibitor pembrolizumab [315]. In addition to being well tolerated, BL-8040 resulted in increased CD8+ T-cells infiltrating the tumor, and concomitant decreases in MDSC and Tregs compared to use of pembrolizumab alone. Although not yet tested in a bone metastasis patient setting, these pre-clinical results are promising as they show tolerability, safety and ability to alter immune cell responses, and together with the added ability to alter DTC dormancy, bone remodeling and angiogenesis, these types of agents may show clinical utility in the treatment of bone metastatic disease.

### 4.4. Integrin-FAK Signaling

Integrins play a major role in cell survival and many cellular processes involved in bone metastasis including bone remodeling, angiogenesis and immune-mediated responses. A number of therapeutic antagonists of integrins have been developed, but generally they fall into two categories: neutralizing antibodies and peptide antagonists. Due to the role of integrins in angiogenesis, cilengitide was developed as an agent to target αvβ3, αvβ5, and α5β1 integrins and is a cyclic pentapeptide antagonist based on the integrin-binding RGD motif of ECM proteins [287]. Its efficacy in inhibition of angiogenesis and tumor growth in preclinical in vivo models has been demonstrated [281,282,283,284], and cilengitide has also been shown to inhibit osteoclast maturation and function [286]. Importantly, cilengitide demonstrated efficacy in the inhibition of breast tumor bone metastasis growth and osteolysis in preclinical models [288,289] and demonstrated reduction in levels of angiogenic factors such as VEGF, PTHrP and RANKL [285]. Despite showing good tolerability, clinical testing of cilengitide did not result in efficacy in numerous reported studies [316] including non-metastatic prostate cancer patients [285]. In a similar manner, antibody antagonists against α5β1 integrin were also well tolerated in patients when added to a chemotherapy treatment regimen of carboplatin and paclitaxel, however added clinical benefit was observed with 33% of advanced lung cancer patients showing partial or objective responses [290]. Abituzumab is another antibody agonist and targets all αv subunit-containing integrins. A phase II trial of abituzumab in prostate cancer patients with progressing bone metastases showed that although no statistically significant effect was observed with respect to progression free survival, abituzumab treatment was associated with a reduction in bone lesion progression compared to placebo treated patients [291]. Although use of integrin antagonists has not shown significant clinical benefit, most studies to date have neglected to examine their use in the bone metastatic setting, and those that have (for example, abituzumab), did not select patients with elevated tumor integrin expression levels whom we can anticipate would show better response to tumor integrin antagonist therapy. Despite literature supporting their utility as a target in preclinical bone metastasis models, it is clear this is not directly translatable to the clinical setting, and perhaps additional studies examining possible compensatory mechanisms by other integrin heterodimers, or studies in immune competent preclinical models are required to determine whether their suggested utility remains possible. As an alternative approach, it may be possible to use integrins as a target to facilitate increased drug uptake in the bone. It has been shown that the β3 integrin subunit is upregulated in bone metastatic breast tumor cells compared to those of primary tumors [317]. Nanoparticles with surface expression of a quinolone non-peptide integrin binding ligand were generated and used to deliver docetaxel pro-drug to preclinical models of breast cancer bone metastases. The authors found that docetaxel was effectively delivered to bone metastases in vivo, resulting in significant reductions in tumor lesion size concomitant with less hepatotoxicity compared to treatment with standard docetaxel. Thus, use of integrin binding to facilitate and enhance drug delivery to bone metastases may be another effective way in which to treat these lesions.

Despite disappointing clinical trial results with agents targeting integrins, clinical studies targeting the downstream effector of integrins, FAK, have also been initiated. Although still in the early stages, phase I/II results have suggested that FAK inhibitors are well tolerated and associated with low grade but clinically manageable adverse events with some modest benefits to clinical outcomes observed [278,279,280]. Although not yet tested in a clinical trial setting designed to treat patients with bone metastases, numerous preclinical studies have shown that inhibition of FAK reduced bone metastatic growth and osteolysis, the latter of which is likely due in part to the suppression of important osteogenic factors such as RANKL [270,271,272]. Inhibition of FAK has also been shown to inhibit tumor growth and angiogenesis in osteosarcoma models [273]. Moreover, FAK inhibitors have been shown to be potent inhibitors of angiogenesis [275], affect osteoclast differentiation and function [274], and regulate the recruitment and activity of immune modulating cells such as Tregs [276,277]. Given its role in almost every facet of bone metastasis progression, investigation into its effective targeting in this setting remains warranted.

### 4.5. The IL-6-STAT3-JAK Signaling Axis

The IL-6-STAT3-JAK pathway has also been shown to be important in tumor progression including that of bone metastases [318]. IL-6, through its ability to signal via the STAT3-JAK pathway, can enhance tumor cell survival [241,242,243,244,245], promote osteoblast and tumor cell production of RANKL and PTHrP to induce osteoclastogenesis [247,248,249], induce angiogenesis [250,251,252,253], and modulate immune cell function [319]. Due to the many effects of IL-6 in the bone metastatic microenvironment, the IL-6-STAT3-JAK axis is a desirable potential target to treat bone metastases. To date, most clinical agents have used antibody antagonists to target IL-6 or IL-6R. A humanized anti-IL-6 antibody, Siltuximab (CNTO328) was shown to be very effective in myeloma patients with increased overall survival in 68% of treated patients when used in combination with chemotherapy [320]. Siltuximab has also been tested in advanced castrate resistant prostate cancer patients with one study suggesting clinical benefit in 30% of patients (by decreased prostate-specific antigen or stable disease responses) [321] and another study showing no effect on clinical outcomes [320,322], however, authors of the latter study note issues in study design that precluded appropriate clinical benefit evaluation. Tocilizumab is a humanized neutralizing antibody to IL-6R which has recently been shown to be well tolerated when given in combination with chemotherapy in ovarian cancer patients [254]. Although this agent has yet to be tested in a bone metastatic patient setting, preclinical studies using similar antibody antagonists to IL-6R were shown to inhibit breast cancer bone metastasis growth and decrease osteolysis concomitant with downregulation of expression of important factors such as VEGF and RANK in preclinical models [246]. In addition to these approaches, many agents targeting STAT3 or JAK kinases downstream of IL-6 receptor binding have been developed and recently reviewed [323]. Of note, the JAK kinase inhibitor AZD4180 was shown to inhibit growth of prostate tumors in a lung metastasis experimental model, however its effects on bone metastases was not evaluated [324]. AZD4180 was also shown to inhibit tumor-associated angiogenesis [325]. Unfortunately, a phase I clinical study of AZD4180 showed unusual dose-limiting toxicities and further studies were halted [326]. Other steric antagonists have been developed, namely FLLL32, which block STAT3 dimerization and activation, and have been shown to inhibit tumor growth and angiogenesis in subcutaneous breast cancer xenograft models [327], however further testing of this molecule clinically has not been reported. One other approach which has shown promising results to date, is the use of anti-sense oligonucleotides (ASO) to target STAT3. An improved class of ASO which are modified by inclusion of 2′-4′ constrained ethyl (cEt)-modified residues were generated targeting STAT3, namely AZD9150 [328]. This agent was shown to potently inhibit tumor growth by 60–90% depending on the tumor type in preclinical in vivo models. Based on these exciting preclinical findings, a phase I clinical trial was initiated, and although dose-limiting thrombocytopenia was observed in many patients, over 40% of patients with refractory lymphoma or lung cancers achieved objective clinical responses [328]. Although not yet tested in bone metastases settings, the demonstrated importance of the IL-6 signaling pathway in bone metastasis progression, and the promising early clinical results in testing agents targeting this pathway suggest it may be a very effective treatment for bone metastatic cancers.

## 5. Conclusions

Bone metastases have serious consequences for patients with cancer and current treatments provide limited palliative support at best. Dissemination of tumor cells to the bone, and the processes by which they alter the bone microenvironment to support their growth and progression involve complex interactions with numerous cell types regulating a variety of biological processes including bone remodeling, angiogenesis and immune cell function. Ideally, therapeutic targets that can block multiple aspects of the bone metastatic process would be most desirable. However, the pleiotropic nature of some targets may limit their utility due to narrow therapeutic windows. Identification of novel effective therapies for bone metastases will likely require their assessment at different stages throughout the bone metastatic process, and their use in combination with other therapeutic modalities such as those currently used as standard of care. Through these detailed investigations, it is hoped we can discover effective therapeutic regimens that can enhance and extend the lives of patients in the near future.

## Figures and Tables

**Figure 1 ijms-22-02911-f001:**
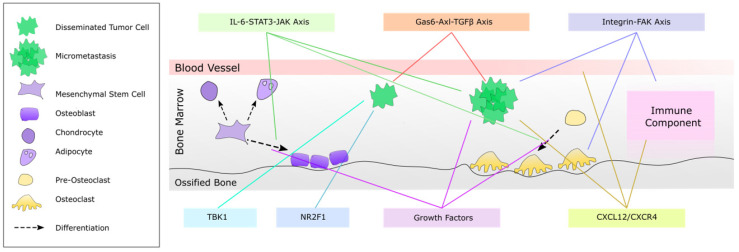
Cell–cell communication pathways as possible targets to prevent development of or treat bone metastasis. Bone metastasis is mediated by many tumor intrinsic factors, but also by cell–cell communication between tumor cells and bone resident cells. Precursor cells residing in the bone marrow can differentiate into immune cells or bone remodeling cell types such as osteoclasts and osteoblasts, which can either inhibit or promote tumor cell growth and progression into a micrometastasis. Cellular processes such as immunosuppression, bone remodeling or angiogenesis can also promote development of macrometastases in the bone, and often share regulatory pathways which could serve as very effective therapeutic targets.

**Table 1 ijms-22-02911-t001:** Potential Therapeutic Targets to Effectively Inhibit Bone Metastasis Development and Progression.

Pathway	Factor (s)	Effect	Potential Therapy
IL-6-STAT3-JAK Axis	IL-6	Tumor cell survival [241,242,243,244,245] Tumor cell expression of osteogenic and angiogenic factors, VEGF and RANK [246] Tumor cell and osteoblast production of RANKL and PTHrP to induce osteoclastogenesis [247,248,249] Angiogenesis [250,251,252,253]	IL-6R antagonists [254,246] Jagged1 inhibitor, 15D11, which decreases osteoblast IL-6 production [129]
GAS6-Axl-TGF-β Axis	GAS6 expression by osteoblasts	Activation of Axl, which is required for TGFβ2-mediated tumor cell dormancy [61]	Propranolol [255] Recombinant GAS6 [256] Agonistic antibodies to Axl [257]
	TGFβR1/2, SMAD3 or downstream CTGF and PAI-1	Tumor cell growth [258] Tumor cell expression of PTHrP [259] Angiogenesis [260,261]	TGFβR1 inhibitor LY2109761 [262] or LY2157299 [263] Specific peptide antagonists of SMAD3 [260,261,264,265,266] TGF-β antibody, fresolimumab [267,268,269]
Integrin-FAK Axis	FAK	Bone metastatic growth and osteolysis [270,271,272,273] Expression of osteogenic and angiogenic factor, RANKL [271] Osteoclast differentiation and function [274] Angiogenesis [275] Recruitment of Tregs [276,277]	FAK inhibitors [270,271,272,273,274,275,276,277,278,279,280]
	Integrin α5β3, α5β5, α5β1	Expression of osteogenic and angiogenic factors, RANKL, VEGF and PTHrP [281,282,283,284,285] Osteoclast maturation and function [286]	Cilengitide [281,282,283,284,285,286,287,288,289] Integrin α5β1 antibody antagonists [290] Abituzumab [291]
TBK1	Stromal cell induction of tumor cell TBK1expression	Chemoresistance via inhibition of mTOR [81]	TBK1 inhibitor, Amlexanox [292]
NR2F1	NR2F1 expression by tumor cells	Tumor cell dormancy [293]	Activators of tumor cell NR2F1, such as combination 5-Aza-C and retinoic acid, to induce or maintain tumor cell dormancy [293]
Growth Factors	VEGF isoforms by osteoblasts, osteoclasts and tumor cells, and P1GF by vasculature	Tumor cell adhesion [153] Angiogenesis [153,155] Osteoblastogenesis [134,135,136] Osteolytic metastasis [140,141]	VEGF inhibitor, cediranib [170,171,172] PEDF-derived peptides [167] or recombinant PEDF [168,169] to suppress angiogenesis, bone resorption and VEGF expression
CXCL12-CXCR4	CXCL12-CXCR4 interaction	Tumor cell proliferation [49,294,295] Metastatic growth [296,297,298] Tumor mass angiogenesis [299] Intratumoral infiltration of MDSC and Tregs [297] Systemic CD8^+^ and CD4^+^ T cells [297]	Plerixafor [296] AMD3465 [297] TC14012 [298]

## Data Availability

Not applicable.

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
