# Peer review of "Targeting Intercellular Communication in the Bone Microenvironment to Prevent Disseminated Tumor Cell Escape from Dormancy and Bone Metastatic Tumor Growth"

_ijms, 2021, doi:10.3390/ijms22062911_

Round 1
Reviewer 1 Report
This well-written review article by Kreps & Addison mainly focuses on the roles of DTCs in the development of bone metastases. Only one point this reviewer would like to suggest is that the words DTCs and/or dormancy should be included in the title.
Author Response
Thank you for the kind comments. We have revised the title to read "Targeting intercellular communication in the bone microenvironment to prevent disseminated tumor cell escape from dormancy and bone metastatic tumor growth. " as recommended by the reviewer.
Reviewer 2 Report
In this manuscript, the authors described the intercellular mechanisms of bone metastasis and discussed the therapeutic potential of targeting intercellular communication between tumor cells and other bone marrow resident cells. The manuscript is well-written and comprehensively covers interesting topics in bone metastasis. The manuscript would help potential readers understand the whole picture of bone metastasis. However, figure 1 looks very busy, and it is hard to read the description in each pathway. I would suggest that the authors summarize that information in a new table in addition to figure 1. Also, osteoblasts are cuboidal cells. It is better to edit the cartoon of osteoblasts in figure 1.
Author Response
Thank you for your kind comments. We have modified Figure 1 to include osteoblasts as cuboidal cells, and have removed the text of the pathways suggested to be targeted. These are now included in Table 1 which has been added to the end of the manuscript document. Pathway labels and the lines connecting them to cell types they can affect remain in place. We have added a line referring to Table 1 in the manuscript on line 542. We hope the figure is now more visually pleasing and the table is sufficiently informative.
Round 2
Reviewer 2 Report
In the revised manuscript, the authors addressed all of my points.